# Effects of exposure to *Rickettsia bellii* on reservoir competence of guinea pigs for *Rickettsia rickettsii*

Lina de Campos Binder[1]/+, Carlos Eduardo Camargo Fanchini[1], Herbert Sousa Soares[1,2], Talita Beck Strabelli dos Santos[1], Sueli Akemi Taniwaki[1], Marcelly Bastos Nassar[3], Andrea Cristina Fogaça[3], Maria Carolina de Azevedo Serpa[1], Matheus Pasini Martins[1], Marcelo Bahia Labruna[1]

[1]Universidade de São Paulo, Faculdade de Medicina Veterinária e Zootecnia, São Paulo, SP, Brasil
[2]Universidade Santo Amaro, Programa de Pós-Graduação em Saúde Única, São Paulo, SP, Brasil
[3]Universidade de São Paulo, Instituto de Ciências Biomédicas, Departamento de Parasitologia, São Paulo, SP, Brasil

**BACKGROUND** *Amblyomma sculptum* is the main vector implicated in the transmission of *Rickettsia rickettsii* in southeastern Brazil, where capybaras are known to be the main amplifying hosts of this pathogen. Interestingly, many areas inhabited by large populations of capybaras and *A. sculptum* remain free of *R. rickettsii*, suggesting that other drivers may be involved in the pathogen perpetuation. *A. sculptum* frequently occurs in sympatry with *Amblyomma dubitatum* ticks, sharing the same hosts with them. Considering that *A. dubitatum* ticks are frequently infected with *Rickettsia bellii*, a possible interference of *R. bellii* in the reservoir competence of capybaras for *R. rickettsii* could explain the heterogeneous distribution of *R. rickettsii* in *A. sculptum* populations in southeastern Brazil.

**OBJECTIVES** The present study aimed to evaluate the effect of primary exposure to *R. bellii*-infected *A. dubitatum* on the reservoir competence for *R. rickettsii* using guinea pigs as experimental models.

**METHODS** Three guinea pigs were infested with *R. bellii*-infected *A. dubitatum* (Group GB), four guinea pigs with uninfected *A. dubitatum* (Group GD), and three guinea pigs were not exposed to *A. dubitatum* (Group GC). After infestation with *A. dubitatum*, all guinea pigs were exposed to one single *R. rickettsii*-infected *A. sculptum* female and then were infested with uninfected *A. sculptum* larvae. After ecdysis, nymphs were tested for detection of spotted fever group rickettsiae.

**FINDINGS** Five out of six guinea pigs from GD and GC groups died, while one out of three guinea pigs in the GB group had a fatal outcome. *R. rickettsii* infection rate among ticks fed on animals from GB group was 21% (17/80), significantly lower than the rate of 54% (60/111) recorded in ticks fed on animals from GD and GC groups.

**MAIN CONCLUSIONS** Prior exposure of guinea pigs to *R. bellii*-infected *A. dubitatum* ticks reduced their reservoir competence for *R. rickettsii*.

Key words: *Amblyomma dubitatum* - *Amblyomma sculptum* - spotted fever - capybara - rickettsiosis

Rocky Mountain spotted fever, also known as Brazilian spotted fever (BSF), is caused by the bacterium *Rickettsia rickettsii*. BSF is the most relevant tick-borne disease in Brazil, where *Amblyomma sculptum* is the main vector implicated in the transmission of this pathogen to humans.[1,2,3,4]

In spite of being a competent vector, *A. sculptum* is partially refractory to *R. rickettsii* infection and less efficient than other tick species in maintaining *R. rickettsii* by transstadial perpetuation and transovarial transmission.[5,6,7] These features suggest that vertical transmission alone is insufficient to perpetuate *R. rickettsii* infection over successive tick generations; thus, horizontal transmission through an amplifying vertebrate host is necessary.[8] Indeed, mathematical models have predicted that after a 90% reduction in amplifying hosts birth rate, *R. rickettsii* infection tends to disappear within two years from *A. sculptum* populations.[9]

Capybaras (*Hydrochoerus hydrochaeris*) are efficient hosts for all *A. sculptum* feeding stages and act as amplifying hosts of *R. rickettsii* for this tick species. [10,11] Since the 1980's, this rodent population has been experiencing rapid growth in southeastern Brazil,[12,13,14] and this has been associated with spatial expansion of BSF occurrence as well as higher incidence rates.[1,8,15] However, it is noteworthy that many areas with high capybara and *A. sculptum* densities remain free of *R. rickettsii* in southeastern Brazil,[16,17,18] indicating that other factors may be involved in the maintenance of the pathogen in vector populations.

Financial support: FAPESP (Grant 2021/06185-9).
+ Corresponding author: binderlina@gmail.com | ⦿ https://orcid.org/0000-0002-2136-7149

**Handling editor:** Ana Carolina Paulo Vicente | ⦿ https://orcid.org/0000-0001-7086-2042

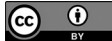

Besides being hosts for *A. sculptum*, capybaras are also primary hosts of *Amblyomma dubitatum*[19,18] and these two ticks occur in sympatry in many areas across southeastern Brazil.[3,18,20,21,22,23] Interestingly, natural infection of *A. dubitatum* with *R. rickettsii* has never been reported, while different studies have described *Rickettsia bellii* infection in *A. dubitatum* populations, often presenting high infection frequencies.[21,24-34]

*Amblyomma sculptum* and *A. dubitatum* were the only ticks found to infest capybaras in anthropised landscapes in southeastern Brazil.[18,35] Nevertheless, Luz et al.[18] noticed that *A. sculptum* greatly outnumbered *A. dubitatum* in BSF-endemic areas, while in BSF-non-endemic areas, abundances of these two species were similar, suggesting a negative correlation between *R. rickettsii* infection and higher *A. dubitatum*/*A. sculptum* abundance ratio.

Spotted fever group (SFG) *Rickettsia* are antigenically related bacteria and exhibit strong cross-reactivity in serological tests. In spite of not being part of SFG, *R. bellii* also presents antigenic cross-reactivity with SFG *Rickettsia*.[36] Considering cross-reactivity between these two bacteria, it would be expected that host immune responses may affect the natural dynamics of these antigenically related organisms. Indeed, several experimental studies have shown that infection with a *Rickettsia* species elicits immune response in hosts reducing or preventing illness and fatal outcomes by a second infection with a pathogenic *Rickettsia*, besides reducing transmission of the pathogen to susceptible ticks.[37,38,39,40,41]

Since *A. sculptum* is not highly efficient in perpetuating *R. rickettsii* infection across generations, amplifying hosts are essential for the pathogen maintenance in vector populations. Therefore, it is plausible that reducing the reservoir competence (ability to transmit an infection to arthropod vectors) of capybaras would affect *R. rickettsii* perpetuation. Considering that capybaras are primary hosts for *A. dubitatum*, and these ticks are frequently infected with *R. bellii*, one can hypothesise that exposure of capybaras to *R. bellii* elicit an immunological response capable of reducing the reservoir competence of capybaras for *R. rickettsii*. Thereby, this study aimed to evaluate if exposure of amplifying hosts to *R. bellii*-infected *A. dubitatum* ticks could reduce or preclude *R. rickettsii* transmission to *A. sculptum* ticks, using guinea pigs as experimental models.

## MATERIALS AND METHODS

*Animals and ticks* - All animals used along the study were kept in the same room, with temperature (23ºC), ventilation and photoperiod (12/12) control. They were maintained in individual cages during the whole experiment, preventing physical contact between them and were fed with a commercial guinea pig pellet diet, fresh grass and water *ad libitum*. Ticks were held in an incubator at 25ºC, 85-90% relative humidity, and total scotophase for off-host developmental stages.

Ticks used in this study derived from colonies maintained in our laboratory by feeding on tick-naïve guinea pigs (*Cavia porcellus*) or tick-naïve rabbits (*Oryctolagus cuniculus*). All the colonies derived from host-seeking adult ticks collected in 2022 in São Paulo municipality, State of São Paulo (-23.557, -46.718). Two colonies of *A. dubitatum* were used, one naturally infected with *R. bellii* and one naturally uninfected, as previously reported.[42] For *A. sculptum*, also two colonies were used, however, one infected with *R. rickettsii* and one uninfected. For establishing the *R. rickettsii*-infected colony, guinea pig organs (liver and lung) infected with *R. rickettsii* strain Itu[3] stored at -80ºC were thawed at room temperature, crushed with sterile phosphate buffered saline (PBS), and the resultant homogenate was inoculated intraperitoneally in guinea pigs, as previously described.[5,7] One day after inoculation, *A. sculptum* larvae were allowed to feed on guinea pigs inside feeding chambers (white cotton sleeves) glued to the animals shaved back. Sleeves were daily opened and detached engorged larvae were removed and taken to an incubator for ecdysis. Thereafter, ticks were fed on tick-naïve guinea pigs.

In order to assess rickettsial infection status of the colonies, flat nymphs and adults were individually tested by different conventional polymerase chain reaction (PCR) protocols. To that end, genomic DNA was extracted from the ticks using the phenol-chloroform-guanidine-isothiocyanate method, as described elsewhere.[43] All samples were initially submitted to a PCR reaction using primers CS-78 and CS-323, targeting a 401 bp fragment of the citrate synthase gene (*gltA*), common in all representatives of the genus *Rickettsia*.[24] DNA samples positive for infection were further tested by two additional protocols: the first one using primers Rr190.70[44] and Rr190.701,[45] which amplifies a 631 bp fragment of the rickettsial 190-kDa outer membrane protein gene (*ompA*) in most of the SFG rickettsiae; and the second one, a *R. bellii*-specific assay, targeting a 338 bp fragment of the *gltA* gene, as previously described.[46]

From the *R. bellii*-infected colony, 100% (48/48) of the *A. dubitatum* individuals were shown to be positive for *R. bellii* and negative for other *Rickettsia* species, while 100% (50/50) of uninfected *A. dubitatum* ticks and 100% (33/33) of uninfected *A. sculptum* ticks were negative for any rickettsial infection. On the other hand, 67% (22/33) of *A. sculptum* ticks exposed to *R. rickettsii* in the larval stage were shown to be infected. Ticks from these four colonies were used in the present study.

*Tick infestations* - Infestation design is summarised in Fig. 1. Ten tick-naïve guinea pigs were allocated in three experimental groups: three guinea pigs previously exposed to *R. bellii*-infected *A. dubitatum* ticks (group GB); four guinea pigs previously exposed to *R. bellii*-uninfected *A. dubitatum* ticks (group GD); and three guinea pigs that were not exposed to *A. dubitatum* ticks (group GC). Guinea pigs from GB and GD groups were initially exposed to *R. bellii*-infected and -uninfected larvae, respectively. Approximately 60 and 120 days after the larval infestation, the same animals were exposed to *A. dubitatum* nymphs and adults, respectively. Guinea pigs from group GC were kept free of ticks along this period.

Thirty days after infestation with *A. dubitatum* adults (GB and GD), guinea pigs from all three groups (GB, GD and GC) were infested by *A. sculptum* ticks. Two feeding chambers were glued to each animal's back. In the first chamber, one single *R. rickettsii*-in-

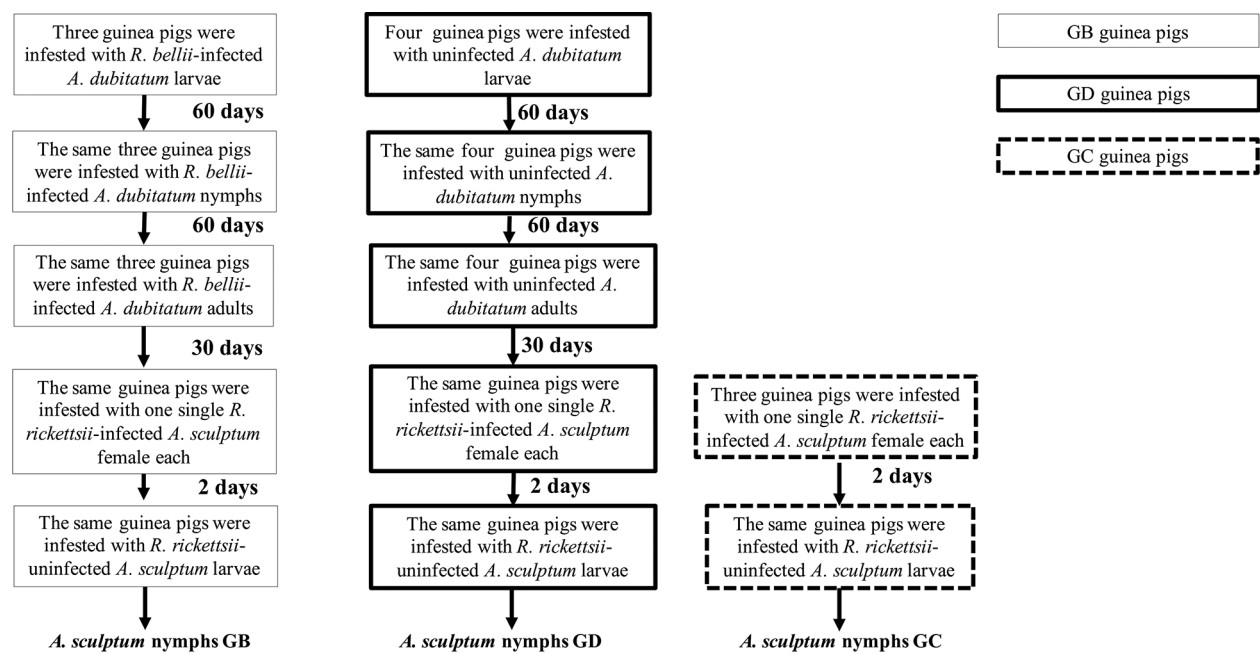

Fig. 1: diagram illustrating experimental procedures with guinea pigs. Groups GB and GD were composed by three and four guinea pigs, respectively, each of them infested by *Amblyomma dubitatum* larvae, nymphs and adults at 60-day intervals. GB served as hosts for *A. dubitatum* ticks infected by *Rickettsia bellii*, whereas GD served as hosts for uninfected *A. dubitatum*. Thirty days later, all animals were exposed to *Rickettsia rickettsii*-infected *Amblyomma sculptum* females (one infected female per guinea pig), and two days later, by uninfected larvae of *A. sculptum*. The GC group, composed of three guinea pigs never exposed to *A. dubitatum* ticks, were submitted to the same protocols of infestation by *A. sculptum* females and larvae.

fected *A. sculptum* female was allowed to feed for two days. In order to infest guinea pigs only with *R. rickettsii*-infected females, previous to the infestation, females had one leg cut off and submitted to DNA extraction by the boiling method.[26] The respective DNA samples were tested by a quantitative polymerase chain reaction (qPCR) protocol (described below). Only females that yielded a positive result in this qPCR protocol were allowed to feed. After two days of feeding, females were detached from the host and submitted to the same qPCR protocol in order to confirm the infection status. One day after the removal of females, uninfected *A. sculptum* larvae (approximately 500 - 1000 larvae/animal) were released in the second chamber and were allowed to feed until repletion. Feeding chambers were opened daily and engorged detached larvae were taken to an incubator for ecdysis. All the procedures were done simultaneously for the three experimental groups.

All animals were monitored for 21 days after each infestation and had their rectal temperature daily measured. Guinea pigs were considered febrile when rectal temperature was > 39.5ºC.[47] In all infestations, approximately 500 µL of blood was collected from the lateral saphenous vein or from the tarsal vein from each guinea pig[48] 0 and 21 day(s) after infestations (DAI). Additionally, febrile animals had blood samples collected daily during the febrile period. Blood samples collected at 0 and 21 DAI were centrifuged, and the respective serum samples were tested for the presence of antibodies against *R. rickettsii* and *R. bellii,* as described below, while blood samples collected during the febrile period were stored at -20ºC for molecular analysis, as described below.

*Serological analyses* - Guinea pig serum samples were individually tested by the indirect immunofluorescence assay (IFA) using a commercial fluorescein isothiocyanate-labelled goat anti-guinea pig IgG (Sigma Diagnostics, Saint Louis, MO) diluted 1:400 and crude antigens derived from *R. rickettsii* strain Taiaçu[49] and *R. bellii* strain Mogi.[49] Each serum was diluted in two-fold increments with PBS from 1:64 to endpoint titre, as previously described.[16] In each slide, a serum previously shown to be non-reactive (negative control) and a known reactive serum (positive control) from a previous study in our laboratory[7] were tested at the 1:64 dilution.

*Rickettsial detection* - After the last infestation with *A. dubitatum* ticks, two males from the *R. bellii*-infected colony, previous fed on each guinea pig from group GB, were dissected and had their salivary glands removed and washed in PBS, as described by Benatti et al.[50] One fragment of the salivary gland was imprinted in a glass slide by gently touching the tissue to the slide in a vertical up-and-down motion. The imprints were left to air dry and then were stained by the Giménez staining.[51] Imprints were evaluated under an optical microscope at a 1000x magnification. The second salivary gland fragment was submitted to DNA extraction using the phenol-chloroform-guanidine-isothiocyanate method.[43] DNA samples were tested by a *R. bellii* specific PCR assay.[46]

DNA samples from *A. sculptum* females used in the infestations were tested by a Taqman qPCR assay targeting a fragment of the citrate synthase gene (*gltA*), using the primers CS-5[52] and CS-6[24] and an internal fluorogenic probe as described by Soares et al.[6] Each reaction

had a positive (*Rickettsia vini* DNA) and a negative control (molecular-grade water).

Blood samples from febrile guinea pigs were submitted to DNA extraction using the DNeasy Blood & Tissue kit (Qiagen, Germany), according to manufacturer instructions. For *R. rickettsii* detection, a probe-based qPCR assay was performed to amplify a gene encoding hypothetical protein A1G_04230 using primers RRi6_F and RRi6_R and an internal fluorogenic probe described by Kato et al.[53] Reactions were performed in a 7500 Real Time PCR Systems apparatus (Applied BioSystems, Foster City, CA). For each reaction, 2.5 μL of DNA template was added to 10 μL of the GoTaq Probe qPCR Master Mix (Promega, Madison, WI), 1.8 μL of each primer at 10 μM, 0.8 μL of probe at 10 μM and molecular-grade water to a final volume of 20 μL. qPCR cycling conditions were 1 cycle of 2 min at 95ºC, followed by 45 cycles of 15 s at 95ºC, 20 s at 55ºC and 30 s at 72ºC.

For quantification of *R. rickettsii* in guinea pig blood samples, plasmids with RRi6_F and RRi6_R amplicons were used to generate a ten-fold dilution standard curve from $10^2$ to $10^6$ copies, which resulted in a linear equation (y = -3.464x + 45.06) with an $R^2$ value = 0.998 and efficiency of 94.383%. The quantification limit of the assay was defined as 100 DNA copies of *R. rickettsii*. Samples and standard curve were run in duplicate. All qPCR results are expressed in bacterial copies per ng of extracted DNA, determined by dividing the calculated copy number per μL by the total DNA concentration (ng/μL) of each sample [quantified in a spectrophotometer (Nanodrop 2000, Thermo Fisher Scientific)].

Approximately 20 days after ecdysis, *A. sculptum* nymphs (fed in the larval stage on febrile guinea pigs) from all the experimental groups underwent individual DNA extraction by the boiling method.[26] DNA samples were initially submitted to a PCR protocol using primers CS-78 and CS-323 targeting a fragment of the citrate synthase gene (*gltA*), common in all representatives of the genus *Rickettsia*.[24] Samples yielding positive results by this first PCR protocol were further tested by a second assay using primers Rr190.70[44] and Rr190.701,[45] which amplifies a fragment of the rickettsial 190-kDa outer membrane protein gene (*ompA*) present in most of the SFG rickettsiae. Nymphs were considered to be infected by *R. rickettsii* only when shown to be positive for both PCR assays. Each reaction had a positive (*Rickettsia vini* DNA) and a negative control (molecular-grade water).

In order to validate DNA extraction, samples that yielded negative results in the rickettsial detection assays were tested by one additional PCR protocol. Tick samples underwent an assay targeting a fragment of the tick mitochondrial 16S rRNA according to Mangold et al.,[54] while blood samples were tested by a PCR targeting a 359 bp fragment of the mitochondrial cytochrome b (*cyt b*) gene from vertebrates, as described elsewhere.[55] Samples with negative results in these PCR protocols were excluded from the study.

*Survey for R. bellii in tick saliva* - Serological analyses showed that no guinea pig from group GB had detectable antibodies against *R. bellii* after exposure to infected *A. dubitatum* ticks (results are described below).

Since *R. bellii* was detected in salivary glands from male *A. dubitatum* ticks fed on all animals from group GB (results are described below), we decided to investigate if *A. dubitatum* ticks secrete *R. bellii* in the saliva. For this purpose, *A. dubitatum* ticks were collected at the exact same locality, where ticks that originated the colonies used in the study were collected.

In order to use only *R. bellii*-infected females, they had one leg cut off and submitted to DNA extraction by the boiling method.[26] The respective DNA samples were tested by a qPCR protocol which amplifies a 338 bp fragment of the *gltA* gene, using primers described by Szabó et al.[46] and an internal fluorogenic probe, as described by Hecht et al.[56] Only five females, those that yielded a positive result in this qPCR protocol, were allowed to feed on a tick-naïve guinea pig along with five *A. dubitatum* males. Females were allowed to feed for six to nine days, until they were partially engorged.

After partial engorgement, females were detached from the host, washed three times in ultrapure water and dried with filter paper. Each female was inoculated with pilocarpine (50mg/ml in 0.7M NaCL solution) using a 12.7 mm x 0.33 mm needle, as described by Esteves et al.[57] Saliva was harvested every five to ten minutes until the female stopped to salivate. One microliter of each saliva sample was dripped onto a slide, left to air dry and then was stained by the Giménez staining.[52] Slides were evaluated under an optical microscope at a 1000x magnification. The remaining saliva samples were stored at -20ºC for molecular analyses. Saliva samples were diluted 1: 2 in Tris-EDTA buffer solution and underwent DNA extraction by the boiling method.[26] For *R. bellii* detection, a specific qPCR assay was performed, as described elsewhere.[56]

*Statistical analyses* - *Rickettsia rickettsii* concentration in blood samples were compared between different experimental groups (GB, GD and GC) by Kruskal-Wallis test and between animals that died and survived by Mann-Whitney test. Proportion of blood samples yielding positive results in qPCR assays for *R. rickettsii* detection as well as *R. rickettsii* infection rate among *A. sculptum* nymphs were compared between the different experimental groups (GB, GD and GC) using Fisher's exact test. Analyses were performed using R[58] and a 5% significance level was assumed.

*Ethics statement* - This study has been approved by the Ethic Committee on Animal Use of the Faculty of Veterinary Medicine and Animal Science, University of São Paulo (CEUA No. 5319211021).

## RESULTS

During all the infestations with *A. dubitatum* ticks, no guinea pig had fever or any other clinical manifestation. Further, no guinea pig seroconverted to *R. bellii* after any of the three consecutive exposures to *R. bellii*-infected *A. dubitatum* ticks (larvae, nymphs and adults). Despite the absence of seroconversion, *R. bellii* DNA was detected in salivary glands of *A. dubitatum* adult ticks that had fed on all guinea pigs from group GB and it was possible to visualise bacteria morphologically compatible with *R. bellii* in their salivary glands (Fig. 2).

In addition, it was possible to visualise bacteria morphologically compatible with *R. bellii* (Fig. 3) and to detect *R. bellii* DNA in saliva samples from all tested females. These coupled results suggest that *A. dubitatum* are able to transmit *R. bellii* to hosts and that animals from the GB group were in fact exposed to *R. bellii* in spite of the absence of a humoral host response.

After exposure to a single *R. rickettsii*-infected tick, all the guinea pigs [except for one guinea pig (C31) from group GC] presented fever and other clinical manifestations compatible with *R. rickettsii* infection, which started between five and 12 DAI. One guinea pig from group GB, three from group GD and two from group GC died 12 to 21 DAI and five to nine days after fever onset (Table I).

It was possible to detect *R. rickettsii* DNA in at least one blood sample from all animals, except for guinea pig C31. Noteworthy, *R. rickettsii* was detected only in one to four timepoints during the whole febrile period (Table I). Proportion of febrile days in which it was possible to detect *R. rickettsii* in blood samples was not different among experimental groups. Bacterial loads were slightly lower among animals from group GB and among animals that survived to infection, but no statistical significance was detected (Fig. 4).

All guinea pigs that had blood samples collected at 21 DAI seroconverted to *R. rickettsii* and *R. bellii*, generally with higher titres for *R. rickettsii* (Table I). Guinea pig C31 (group GC), which was the only one

to remain afebrile, was also the only one that did not seroconvert, indicating that this animal was in fact not exposed to the pathogen. For this reason, it was not considered in further analyses.

From the *A. sculptum* nymphs that fed as larvae on guinea pigs during their febrile period, 191 nymphs underwent molecular analyses, which showed that 77 (40%) of them were infected by *R. rickettsii*. When evaluating the different experimental groups separately, ticks fed on GB guinea pigs showed 21% (17/80) infection rate, while ticks from groups GD and GC had infection rates of 50% (49/97) and 79% (11/14), respectively. When analysing ticks from GD and GC together, an infection rate of 54% (60/111) was observed. *R. rickettsii* infection rate in GB ticks was significantly lower than the infection rates recorded on the GD and GC groups (both when analysed separately or together) (Table II).

## DISCUSSION

In the present study, the reservoir competence of guinea pigs for *R. rickettsii* was evaluated by comparing the infection rates of *A. sculptum* nymphs after feeding as larvae on three groups of *R. rickettsii*-infected guinea pigs: one group previously exposed to *R. bellii*-infected *A. dubitatum* ticks (GB), one group previously exposed to uninfected *A. dubitatum* ticks (GD), and one group never previously exposed to *A. dubitatum* (GC). As the GB *A. sculptum* nymphs had a significantly lower *R.*

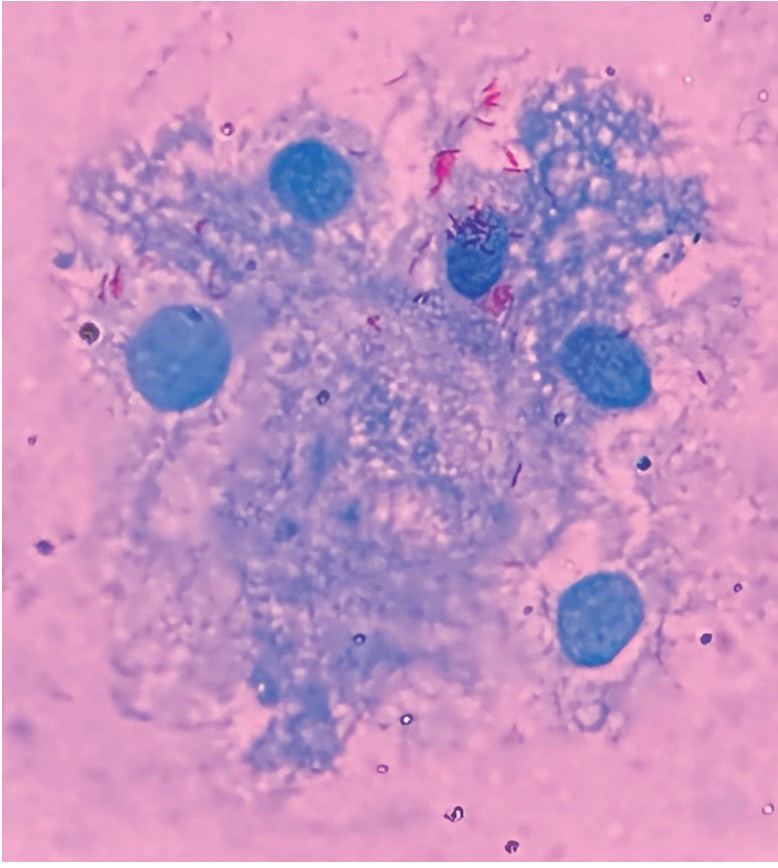

Fig. 2: optical microscopy analysis of the salivary glands of *Amblyomma dubitatum* ticks infected with *Rickettsia bellii* (Giménez staining, 1000x original magnification).

*rickettsii*-infection rate than the GD and GC nymphs, it was demonstrated under experimental conditions that exposure to *R. bellii*-infected *A. dubitatum* ticks reduced the reservoir competence of guinea pigs for *R. rickettsii*.

After being exposed to *R. rickettsii*-infected females, all guinea pigs (except for C31) presented clinical illness and seroconverted to *R. rickettsii* or died during acute infection. Although it was possible to detect *R. rickettsii* DNA in blood samples from all animals, it was not possible to detect the pathogen in all blood samples collected daily during the febrile period. *R. rickettsii* infects endothelial cells and has a very low concentration in the bloodstream, which makes blood qPCR an unsuitable assay for *R. rickettsii* diagnosis, as reported by several other studies, in which blood samples from infected animals yielded negative results in blood qPCR assays.[10,11,59,60]

Santos et al.[61] showed that among BSF human cases, *R. rickettsii* DNA was detected in serum samples from all patients that had fatal outcomes, while in nonfatal cases, qPCR yielded positive results only for one third of the samples. Fatal outcomes are probably related to a vaster endothelial injury leading to severe vascular alterations and therefore higher *R. rickettsii* concentration in the blood. Interestingly, herein no statistical difference was observed in *R. rickettsii* load between animals that died or survived infection, even though a slightly higher load was recorded in animals that had fatal outcomes.

Even though guinea pigs from group GB showed a slightly lower *R. rickettsii* load, no statistical difference was observed among different experimental groups, precluding association between qPCR results and reservoir competence, assessed by the number of ticks becoming infected after blood feeding. This could be due to our small sample of guinea pigs or to an unsuitability of blood qPCR to assess hosts infectivity. In this regard it is important to keep in mind that molecular assays reflect one single moment in time when the respective blood sample was collected, while ticks remain attached ingesting blood for days. Further, it is not possible to affirm that *R. rickettsii* DNA detected by qPCR corresponds to viable bacteria, precluding inferences on infectivity.

Guinea pig C31 was not exposed to *R. rickettsii* as verified by the absence of seroconversion. This was unexpected, since the *A. sculptum* female fed on this guinea pig was shown to be infected. One possible reason for the absence of *R. rickettsii* transmission is the lack of time for the female to transmit the agent. Magalhães[62] demonstrated that *A. sculptum* ticks may need 36 h to efficiently transmit *R. rickettsii* to a susceptible host. In our study, feeding chambers were inspected two times a day until the female was seen to be attached to the host. Forty-eight hours after attachment, females were removed from the chambers. Nevertheless, during this 48 h period, feeding chambers were not opened, being not possible to state that ticks were continuously attached to the host during this entire period.

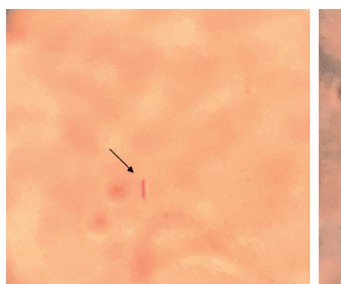 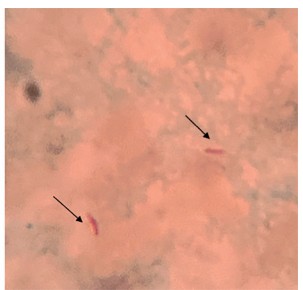

Fig. 3: optical microscopy analysis of saliva samples of *Amblyomma dubitatum* ticks infected with *Rickettsia bellii* (Giménez staining, 1000x original magnification).

TABLE I

Clinical and serological data of guinea pigs infested with *Rickettsia rickettsii*-infected *Amblyomma sculptum* females

| Experimental group | Guinea pig ID | Endpoint titre to *R. bellii*[a] | Endpoint titer to *R. rickettsii*[a] | Febrile period (DAI) | Death (DAI) | *R. rickettsii* DNA detected in blood (DAI) | No. of bacteria/ng of extracted DNA* |
|---|---|---|---|---|---|---|---|
| GB | C10 | 4096 | 16184 | 7 - 11 | NO | 8, 11 | 10.0 (7.0-12.9) |
| | C11 | NA | NA | 7 - 11 | 13 | 10, 11 | 235.7 (26.9-444.5) |
| | C12 | 4096 | 8192 | 6 - 14 | NO | 9, 11, 12, 14 | 35.1 (4.1-129.0) |
| GD | C16 | 4096 | 4096 | 7 - 13 | NO | 12, 13 | 57.8 (44.6-70.9) |
| | C17 | 512 | 1024 | 12 - 18 | 21 | 14, 15 | 81.8 (29-134.6) |
| | C18 | NA | NA | 7 - 10 | 13 | 9 | 6.4 |
| | C19 | NA | NA | 6 - 10 | 12 | 8, 10 | 285.2 (34.2-536.2) |
| GC | C30 | NA | NA | 5 - 8 | 10 | 5, 6, 7, 8 | 42.0 (2.6-146.7) |
| | C31 | <64 | <64 | NO | NO | NO | |
| | C32 | NA | NA | 6 - 10 | 12 | 7, 9 | 58.45 (13.3-103.6) |

GB: guinea pigs previously exposed to *Rickettsia bellii*-infected *Amblyomma dubitatum* ticks; GD: guinea pigs previously exposed to *R. bellii*-uninfected *A. dubitatum* ticks; GC: guinea pigs not exposed to *A. dubitatum* ticks; DAI: days after infestation with *R. rickettsii*-infected tick; NA: not analysed (animal died before 21 DAI); NO: not observed. *a*: Endpoint titre to *R. rickettsii* or *R. bellii* at 21 DAI; all guinea pigs were seronegative (endpoint titer < 64) at 0 DAI; *median (range).

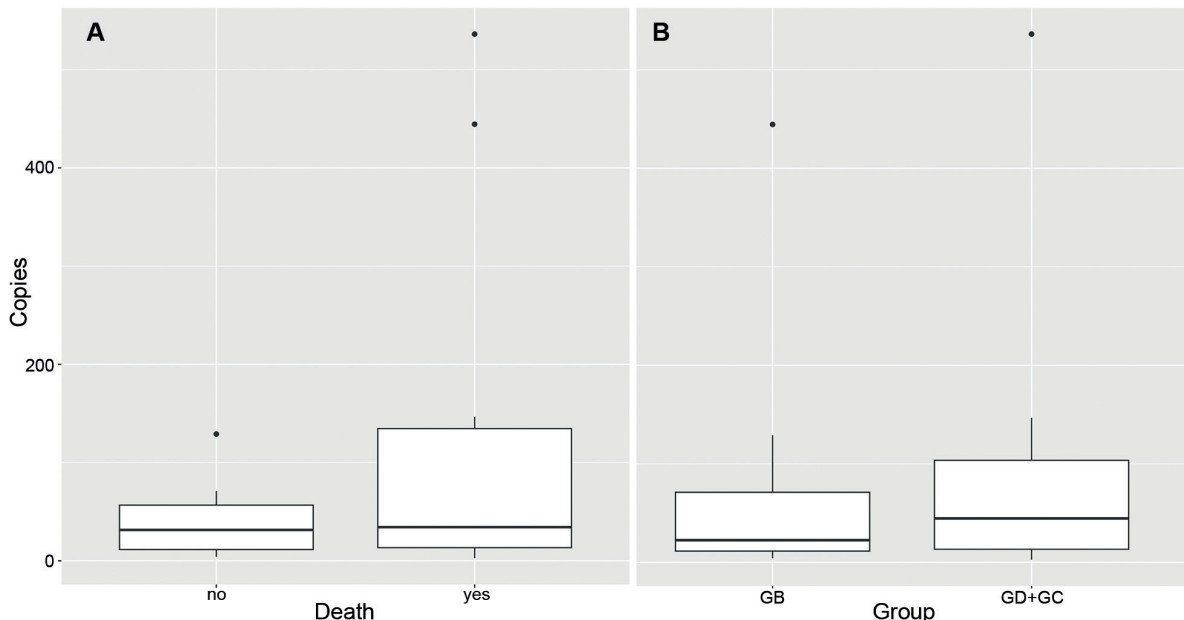

Fig. 4: (A) *Rickettsia rickettsii* load in blood samples (No. of *R. rickettsii* genome copies/ng of extracted DNA) from guinea pigs that died after infection and guinea pigs that survived infection with *R. rickettsii*. (B) *R. rickettsii* load in blood samples (No. of *R. rickettsii* genome copies/ng of extracted DNA) from guinea pigs infested with *R. rickettsii*-infected *Amblyomma sculptum* females. GB: guinea pigs previously exposed to *Rickettsia bellii*-infected *Amblyomma dubitatum* ticks; GD: guinea pigs previously exposed to *R. bellii*-uninfected *A. dubitatum* ticks; and GC: guinea pigs not exposed to *A. dubitatum* ticks.

TABLE II

*Rickettsia rickettsii* infection rates among *Amblyomma sculptum* unfed nymphs that had fed as larvae on guinea pigs previously exposed to *R. rickettsii*-infected *A. sculptum* females

| Experimental group | Guinea pig ID | Infection rate (No. of infected nymphs/No. of tested nymphs)[*][+] |
|---|---|---|
| GB | C10 | NA |
| | C11 | NA |
| | C12 | 21% (17/80) |
| | Total | 21% (17/80)[a] |
| GD | C16 | 0% (0/4) |
| | C17 | NA |
| | C18 | 41% (7/17) |
| | C19 | 55% (42/76) |
| | Total | 50% (49/97)[b] |
| GC | C30 | NA |
| | C32 | 79% (11/14) |
| | Total | 79% (11/14)[b] |
| GD + GC | Total | 54% (60/111)[b] |

GB: guinea pigs previously exposed to *Rickettsia bellii*-infected *Amblyomma dubitatum* ticks; GD: guinea pigs previously exposed to *R. bellii*-uninfected *A. dubitatum* ticks; GC: guinea pigs not exposed to *A. dubitatum* ticks; NA: not analysed (none engorged larva molted to nymphal stage after feeding on this guinea pig during its febrile period); *Infection rate was determined based on a polymerase chain reaction (PCR) protocol targeting a fragment of the *ompA* gene; [+]values followed by different superscript letters in the same column are significantly different (p < 0.05).

Despite being repeatedly exposed to *R. bellii*, which was confirmed by detection of this bacterium in salivary glands of ticks fed on all animals, no guinea pig from group GB seroconverted to *R. bellii* prior to exposure to *R. rickettsii*. This was also noticed by other studies[29,42] and suggest that *R. bellii* exposure through tick bite is not able to elicit a humoral response in guinea pigs. These results are quite surprising, since many serosurveys have detected different wild and domestic animals with IgG titers for *R. bellii* higher than for other *Rickettsia* species, suggesting a homologous reaction to *R. bellii*.[18,63,64] One hypothesis is that these seropositive animals were ex-

posed to *R. bellii* through a different infection route (*e.g.*, ingestion of infected ticks), which enabled a humoral response due to exposure to a higher bacterial load. Another hypothesis is that antibodies detected in these studies do not derive from a previous exposure to *R. bellii*, but to another agent that cross-reacts with this bacterium.

In the study by Pacheco et al.,[16] guinea pigs were inoculated intraperitoneally with different species of *Rickettsia* (*R. bellii*, *Rickettsia canadensis*, *Rickettsia monteiroi* and *R. rickettsii*). While guinea pigs inoculated with *R. bellii* had titres to the homologous antigen ranging from 256 to 1024, animals inoculated with other *Rickettsia* showed much higher homologous titres, ranging from 2048 to 32768. This low humoral response to *R. bellii*, even after exposure to a high number of bacteria via intraperitoneal inoculation, may indicate a low ability of *R. bellii* to elicit antibodies production. Further studies should be performed to elucidate these findings and to ascertain whether IFA is a suitable assay for determining a previous exposure to *R. bellii* via tick bite, or not.

Since *Rickettsia* spp. are obligate intracellular bacteria, cell-mediated immunity, instead of humoral immunity, is expected to be the main immune response involved in their clearance, which is corroborated by studies that evaluated passive transfer of T cells in infections by different species of *Rickettsia*.[65,66,67,68] Further, it was observed that mice lacking CD8+ T cells showed reduced survival and increased bacterial load after infection with *Rickettsia conorii* and enhanced susceptibility to fatal outcomes after infection with *Rickettsia australis*.[67,68] Thus, absence of detectable IgG against *R. bellii* does not necessarily mean that no effective adaptive immunity was induced. On the other hand, there is evidences that humoral response, specially antibodies to epitopes of OmpB, also play a relevant role in reinfections by *Rickettsia*, reducing the severity of the disease.[69,70,71] This could suggest that the adaptive immunity elicited by *R. bellii* may not be as effective as immune responses induced by SFG *Rickettsia* in protecting hosts to a second infection with a more pathogenic *Rickettsia*.

Indeed, exposure to *R. bellii* did not prevent clinical illness or fatal outcomes after infection by *R. rickettsii*, however, the lethality rate was lower in group GB (1/3) than in the other experimental groups (5/6). These results are different from other studies in which exposure to different *Rickettsia* fully prevented death due to *R. rickettsii* infection.[37,38,39,41] These studies, however, used *Rickettsia* species phylogenetically closer to *R. rickettsii* than *R. bellii*, possibly inducing a stronger cross-reaction to *R. rickettsii* than the one observed in this study and, therefore, being more effective in protecting the hosts from severe illness and death.

Even though exposure to *R. bellii*-infected ticks did not fully protect animals from illness nor prevented *A. sculptum* larvae from being infected by *R. rickettsii*, guinea pigs from group GB (previously exposed to *R. bellii*-infected ticks) were less efficient in infecting *A. sculptum* larvae, when compared to guinea pigs from the other experimental groups (not previously exposed to *R. bellii*). Considering that *A. sculptum* is not very efficient in maintaining *R. rickettsii* across successive generations, reduction in host reservoir competence would probably have a great impact on the perpetuation of this pathogen. Nevertheless, since *A. sculptum* larvae were fed on a few animals, it is not possible to completely rule out the hypothesis that the difference in reservoir competence between experimental groups is, in part, due to natural variability among individuals rather than to previous exposure to *R. bellii*.

Binder et al.[42] showed that *A. dubitatum* ticks were competent vectors for *R. rickettsii* and also able to transmit the pathogen transovarially, suggesting that these ticks may be important in the enzootic cycle of *R. rickettsii*. On the other hand, the same study noticed that *A. dubitatum* females coinfected by *R. rickettsii* and *R. bellii* generated a low number of viable larvae. Taken together, these results suggest that *R. bellii* could negatively impact the perpetuation of *R. rickettsii* among *A. sculptum* populations by two mechanisms: first, by reducing capybaras' reservoir competence for *R. rickettsii*, and second, by minimising the role of *A. dubitatum* in *R. rickettsii* enzootic cycle due to the negative impact of superinfection on tick viability with a consequent reduction in the population of *R. rickettsii*-infected *A. dubitatum* ticks. These mechanisms could explain the association between a higher *A. dubitatum*/*A. sculptum* abundance ratio and the absence of *R. rickettsii* noticed by Luz et al.,[18] being important drivers associated to spatial heterogeneity of *R. rickettsii*-infected *A. sculptum* populations within southeastern Brazil.

In conclusion, our results demonstrated that prior exposure of guinea pigs to *R. bellii* reduced their reservoir competence for *R. rickettsii*. Under natural conditions, this kind of interference among capybaras may significantly influence the long-term persistence of *R. rickettsii* in *A. sculptum* populations, a condition yet to be confirmed in the capybara model. Therefore, *R. bellii* may represent a promising strategy for disrupting the amplifying role of capybaras in the *R. rickettsii* transmission cycle.

## AUTHORS' CONTRIBUTION

LCB and MBL - study conception and design; LCB, HSS, TBSS, CECF and MPM - experimental infection trials; LCB, CECF, SAT, MBN, MCAS and ACF - molecular and serological trials; LCB and MBL - formal analysis; LCB - writing original draft. All authors critically revised the work and approved the final manuscript.

## DATA AVAILABILITY

The contents underlying the research text are included in the manuscript.

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

# OPEN PEER REVIEW

Memórias do IOC thanks the anonymous reviewers for their contribution to the peer review of this work.

## FIRST REVIEW ROUND

REVIEWERS' COMMENTS

### REVIEWER #1

The manuscript titled "Effects of exposure to Rickettsia bellii on reservoir competence of guinea pigs for Rickettsia rickettsii" contains relevant information, and I recommend it for publication. However, I suggest that the authors make some minor changes, if they deem it appropriate.

After reading about the experimental design, I felt that an experimental group exposed to R. bellii through intraperitoneal inoculation was missing, so that we would have a group with anti-R. bellii antibodies, since in non-endemic areas of BSF, the final titres in capybaras were significantly higher for R. bellii, according to Luz et al. (2019). My interest would be in the response regarding the behaviour of both the host reservoir and the Amblyomma sculptum tick population in this group "R. bellii-infected guinea pig via intraperitoneal route", which was not carried out in this experiment. Is there any reason  not to include this group in the experiment, even if it does not mimic what happens in the natural environment, besides the clarification related to the work of Pacheco et al. (2007)? As also discussed in the manuscript, capybaras could ingest R. bellii-infected ticks while feeding and for this reason would have anti-R. bellii antibodies. I apologise if I was not clear in these questions: I just wanted to understand the absence of this group in the experiment.

Another issue that I consider significant is that the authors reported that the guinea pigs were exposed to R. bellii because found this rickettsia in the salivary gland A. dubitatum tick, ok? The question is: were the animals really exposed to R. bellii, since they did not develop anti-R. bellii antibodies using the IFA technique? And if there is any "unknown factors inherent to A. dubitatum tick" that prevents the entry of this bacterium into guinea pigs during the tick's blood meal? In this case, it is likely that the guinea pigs were not exposed to R. bellii, right?

Materials and Methods
Pag. 5; line 100: If possible, change "Ticks" to "Animals and Ticks"

Results
Pag. 11; lines 257-258: "These coupled results suggest that animals from the GB group were in fact exposed to R. bellii in spite of the absence of a humoral host response". Please, review the work of Sakay et al. (2014) which states that "These results suggest that R. bellii was not tick-transmitted to CG guinea pigs. Guinea pigs exposed to ticks infected by both R. rickettsii and R. bellii generally elicited endpoint titers much higher to R. rickettsii than to R. bellii, indicating that reactions to the later agent were a result of cross-reactions with anti-R. rickettsii antibodies". The question is: Does R. bellii-infected A. dubitatum tick transmit the bacterium to guinea pigs or not?

Pag. 11; lines 273-275: In this way, we can consider that all animals in the GB group were not exposed to rickettsia, correct?

Discussion
Pag. 11; lines 331-332: Or, for some unknown reason, the tick failed to transmit the bacteria during the blood meal, even though it was detected by direct methods in the salivary glands.

### REVIEWER #2

The article by Binder and colleagues describes a study aimed at exploring an interesting phenomenon regarding the ecology of Rickettsia rickettsii in Brazil, where Rocky Mountain spotted fever (also known as Brazilian spotted fever) is endemic.  In general, the manuscript is well written.

The overall experimental design is good, but this design is greatly limited by the very few animals.  Although the total number of ticks is great, the number of animals for which the experiment was conducted was small (10 guinea pigs separated into 3 experimental groups) with lack of justification for these small numbers via power analysis.  As an outbred animal model, one can expect more variation limiting validity of results with the few animals used.  In addition, one animal (a control) was not analyzed due to the assumption that it was not infected (it should be noted that even ~ 12-15% of Hartley strain guinea pigs are completely resistant to infection with Rickettsia rickettsii).  Although there is a statistical difference between the infection rate of nymphs, these nymphs were collected from very few animals, which questions if the comparison of these results to the situation in nature is valid.

The length of the manuscript is not commensurate for the amount and type of data presented. Both the introduction, discussion, and reference lists should be consolidated.

Lines 175 – 176:  Please provide the dilution of the secondary antibody.

Line 180: Comment. A dilution of 1:64 is quite low for a positive control. Would usually expect a known positive to require a higher dilution for optimized viewing.  If indeed this low dilution was required, it brings to question the quality of antigen used for the IFA analysis.

Llines 338 – 339: Seropositivity through ingestion of Rickettsia-infected ticks rather than tick bites?  Seems unlikely.

## AUTHORS' RESPONSE TO THE REVIEWERS

We thank the reviewers for their valuable suggestions and important criticisms. Below are the responses to each of the comments.

We would also like reviewers to be aware that we conducted an additional experiment to address some of the questions raised, which is described in the new version of the manuscript (L. 237 – 262; L. 285 - 289).

Reviewer:1

The manuscript titled "Effects of exposure to Rickettsia bellii on reservoir competence of guinea pigs for Rickettsia rickettsii" contains relevant information, and I recommend it for publication. However, I suggest that the authors make some minor changes, if they deem it appropriate.

After reading about the experimental design, I felt that an experimental group exposed to R. bellii through intraperitoneal inoculation was missing, so that we would have a group with anti-R. bellii antibodies, since in non-endemic areas of BSF, the final titres in capybaras were significantly higher for R. bellii, according to Luz et al. (2019). My interest would be in the response regarding the behaviour of both the host reservoir and the Amblyomma sculptum tick population in this group "R. bellii-infected guinea pig via intraperitoneal route", which was not carried out in this experiment. Is there any reason not to include this group in the experiment, even if it does not mimic what happens in the natural environment, besides the clarification related to the work of Pacheco et al. (2007)? As also discussed in the manuscript, capybaras could ingest R. bellii-infected ticks while feeding and for this reason would have anti-R. bellii antibodies. I apologise if I was not clear in these questions: I just wanted to understand the absence of this group in the experiment.

RESPONSE: Thanks for this interesting and relevant comment. Indeed, if we were to repeat this experiment today, we would definitely include a group with intraperitoneal inoculation of R. bellii. However, at the time when we planned this study, we were interested in mimicking a natural exposure to R. bellii via tick bite, because we thought that this would be the most common natural infection route. At that time, we did not imagine that guinea pigs would not develop a humoral response against R. bellii. Although it was published earlier, the other study in which our team observed this behavior (Binder et al., 2025) was carried out several months after this one. That is why we did not include other experimental groups in the study (e.g. a group exposed to R. bellii via tick ingestion).

Another issue that I consider significant is that the authors reported that the guinea pigs were exposed to R. bellii because found this rickettsia in the salivary gland A. dubitatum tick, ok? The question is: were the animals really exposed to R. bellii, since they did not develop anti-R. bellii antibodies using the IFA technique? And if there is any "unknown factors inherent to A. dubitatum tick" that prevents the entry of this bacterium into guinea pigs during the tick's blood meal? In this case, it is likely that the guinea pigs were not exposed to R. bellii, right?

RESPONSE: We thought that detecting R. bellii in salivary glands was enough to state that this bacterium was inoculated in guinea pigs, but after this valuable questioning, we decided to verify whether R. bellii is in fact secreted in the saliva. Thus, we conducted an additional experiment, which is described in the new version of this manuscript, in which we were able to detect R. bellii in the saliva of infected A. dubitatum, corroborating the hypothesis that guinea pigs were indeed exposed to R. bellii, despite of the absence of detectable humoral response (L. 237 – 262; L. 285 - 289).

Materials and Methods
Pag. 5; line 100: If possible, change "Ticks" to "Animals and Ticks"
RESPONSE: The word "animals" was added.

Results
Pag. 11; lines 257-258: "These coupled results suggest that animals from the GB group were in fact exposed to R. bellii in spite of the absence of a humoral host response". Please, review the work of Sakay et al. (2014) which states that "These results suggest that R. bellii was not tick-transmitted to CG guinea pigs. Guinea pigs exposed to ticks infected by both R. rickettsii and R. bellii generally elicited endpoint titers much higher to R. rickettsii than to R. bellii, indicating that reactions to the later agent were a result of cross-reactions with anti-R. rickettsii antibodies". The question is: Does R. bellii-infected A. dubitatum tick transmit the bacterium to guinea pigs or not?

RESPONSE: Sakai et al (2014) did not attempt to detect R. bellli neither in the salivary glands or in the saliva of A. dubitatum, so, as there was no detectable humoral response, they assumed that R. bellii was not transmitted to guinea pigs. In this study we were able to better investigate the behavior of R. bellii in A. dubitatum ticks and it seems very unlikely that R. bellii is not transmitted to guinea pigs, since the bacterium was detected in salivary glands and saliva in 100% of the ticks that were tested.

Pag. 11; lines 273-275: In this way, we can consider that all animals in the GB group were not exposed to rickettsia, correct?

RESPONSE: In this paragraph we are referring to the absence of humoral response and clinical manifestations in animal C31 after exposure to R. rickettsii. All animals of the other groups had clinical signs and seroconverted after exposure to R. rickettsii. The absence of humoral response after exposure to R. bellii is discussed along the manuscript.

Discussion

Pag. 11; lines 331-332: Or, for some unknown reason, the tick failed to transmit the bacteria during the blood meal, even though it was detected by direct methods in the salivary glands.

RESPONSE: Since we were able to show that A. dubitatum secretes R. bellii in the saliva, we do not see a plausible reason for this bacterium not being transmitted to guinea pigs. Additionally, our findings suggest a different immunological behavior among guinea pigs previously exposed to R. bellii, which reinforces that these animals were indeed infected by this bacterium.

Reviewer 2:

The article by Binder and colleagues describes a study aimed at exploring an interesting phenomenon regarding the ecology of Rickettsia rickettsii in Brazil, where Rocky Mountain spotted fever (also known as Brazilian spotted fever) is endemic. In general, the manuscript is well written.

The overall experimental design is good, but this design is greatly limited by the very few animals. Although the total number of ticks is great, the number of animals for which the experiment was conducted was small (10 guinea pigs separated into 3 experimental groups) with lack of justification for these small numbers via power analysis. As an outbred animal model, one can expect more variation limiting validity of results with the few animals used. In addition, one animal (a control) was not analyzed due to the assumption that it was not infected (it should be noted that even ~ 12-15% of Hartley strain guinea pigs are completely resistant to infection with Rickettsia rickettsii). Although there is a statistical difference between the infection rate of nymphs, these nymphs were collected from very few animals, which questions if the comparison of these results to the situation in nature is valid.

RESPONSE: The small number of animals is unfortunately a limitation of the study. It is difficult nowadays to justify using a larger number of animals for this type of experiment. Rivas et al. (2015), for example, investigated protective immunity against R. rickettsii elicited by R. amblyommatis using only eight guinea pigs, while Levin et al. (2014) assessed effects of heterologous immunization in reservoir competence of dogs using only six animals. Furthermore, infection by R. rickettsii causes substantial distress in guinea pigs and this was a fairly long experiment, with animals being kept in individual cages without social interaction for several months, what makes even more difficult to advocate for the use of larger numbers of animals.

The length of the manuscript is not commensurate for the amount and type of data presented. Both the introduction, discussion, and reference lists should be consolidated.

RESPONSE: Reference list was consolidated. On the other hand, the Introduction was formulated to contextualize the application of the experiments used in this study, while the Discussion section discusses precisely the application of these results in the ecology of Brazilian spotted fever. Unfortunately, we were unable to determine what could be omitted as being of lesser importance. If this reviewer can suggest this in a more objective manner, we would be grateful.

Lines 175 – 176: Please provide the dilution of the secondary antibody.

RESPONSE: Dilution of the secondary antibody is now described.

Line 180: Comment. A dilution of 1:64 is quite low for a positive control. Would usually expect a known positive to require a higher dilution for optimized viewing. If indeed this low dilution was required, it brings to question the quality of antigen used for the IFA analysis.

RESPONSE: Although we used a dilution of 1:64 for the positive control serum, this serum had an endpoint titer higher than 1024, as obtained from guinea pigs previously inoculated with R. rickettsii in our laboratory. We have added the reference of Gerardi et al. (2019) for this positive control serum in the text. We used a dilution of 1:64 just because it was the cut-off dilution for the test serum samples. But indeed, this same positive control serum could be used at higher dilutions (e.g., 1:512 or 1:1024) without any interference in our serological analyses.

Lines 338 – 339: Seropositivity through ingestion of Rickettsia-infected ticks rather than tick bites? Seems unlikely

RESPONSE: It is just a hypothesis, not a conclusion or a statement.

## SECOND REVIEW ROUND

<div align="right">REVIEWERS' COMMENTS</div>

**REVIEWER #1**

The authors included a microscopy image of tick saliva to demonstrate the presence of bacteria (R. belli) and, consequently, transmission, but they do not mention the bacterial load and viability in the saliva (I found it quite low—only two bacteria in the image). Were the bacteria viable and actually capable of establishing an infection and, "somehow," preventing infection by R. rickettsii? Or is referee 2 correct in mentioning that resistance to infection can occur in these animals?

**REVIEWER #2**

The manuscript addresses this reviewer's minor comments, but it is unresponsive to comments regarding animal numbers.  Their justification based on other small studies with different experimental designs is not an appropriate approach to study design.  Distress is here used as a convenient rationale, rather than a scientifically based one.  No additional discussion about these weaknesses of low animal numbers is provided in the discussion.  At the very least, the latter is required.

<div align="right">AUTHORS' RESPONSE TO THE REVIEWERS</div>

We thank the reviewers for their valuable suggestions and important criticisms. Below are the responses to each of the comments.

Reviewer 1:
The authors included a microscopy image of tick saliva to demonstrate the presence of bacteria (R. belli) and, consequently, transmission, but they do not mention the bacterial load and viability in the saliva (I found it quite low—only two bacteria in the image). Were the bacteria viable and actually capable of establishing an infection and, "somehow," preventing infection by R. rickettsii? Or is referee 2 correct in mentioning that resistance to infection can occur in these animals?

Response:
In the previous version of the manuscript, this reviewer had made the following question: "In this case, it is likely that the guinea pigs were not exposed to R. bellii, right?" With our new procedure to demonstrate R. bellii organisms in the saliva of A. dubitatum, we can confirm that guinea pigs were exposed to R. bellii after been infested with ticks naturally infected with this agent. Since these A. dubitatum ticks were naturally infected with R. bellii, and were shown to maintain this infection through successive life stages and generations in our laboratory, we see no reasonable evidence that the bacteria were not viable.

Regarding our new figure of the manuscript, we see two bacteria at a magnification of 1000x in a single field. In 1µl of saliva (volume of saliva observed under the microscope), we observed more than 20 bacteria in all samples. Considering that ticks salivate for days on the host, we do not believe this to be a negligible amount of bacteria. We are unaware of any studies that have quantified Rickettsia in tick saliva, so any results we obtained would not be comparable to other studies or other Rickettsia species. Regardless, we recall that the procedure for demonstrating the presence of rickettsia in saliva was to answer this reviewer's final question: "In this case, it is likely that the guinea pigs were not exposed to R. bellii, right?". Obviously, quantification of rickettsia (bacteria load) in tick saliva is far beyond of the scope of this manuscript.

Please note that in our Discussion, we do not argue that R. bellii was able to establish an infection in the host, but only that it could have induced a cellular immune response, based on the general lower mortality of guinea pigs after being exposed to R. bellii-infected ticks. We would like to reinforce that guinea pigs are the animal model of choice for R. rickettsii infection, and we are unaware of any studies showing that any of these animals are resistant to highly virulent strains of R. rickettsii.  Just to point out, we used a strain of R. rickettsii that is very virulent for R. rickettsii, as demonstrated in previous studies of experimental infection with this strain in guinea pigs (Soares et al. 2012, Gerardi et al. 2019).

Reviewer 2:
The manuscript addresses this reviewer's minor comments, but it is unresponsive to comments regarding animal numbers.  Their justification based on other small studies with different experimental designs is not an appropriate approach to study design.  Distress is here used as a convenient rationale, rather than a scientifically based one.  No additional discussion about these weaknesses of low animal numbers is provided in the discussion.  At the very least, the latter is required.

Response: A discussion about limitations of using a small sample of animals was included in the manuscript (L 411 - 414).

## THIRD REVIEW ROUND

REVIEWERS' COMMENTS

**REVIEWER #1**

No comments.

**REVIEWER #2**

No comments.

