## [Reviewer Report · FIRST REVIEW ROUND - REVIEWERS COMMENTS]

## REVIEWER #1

The manuscript titled “Effects of exposure to Rickettsia bellii on reservoir competence of guinea pigs for Rickettsia rickettsii” contains relevant information, and I recommend it for publication. However, I suggest that the authors make some minor changes, if they deem it appropriate.

After reading about the experimental design, I felt that an experimental group exposed to R. bellii through intraperitoneal inoculation was missing, so that we would have a group with anti-R. bellii antibodies, since in non-endemic areas of BSF, the final titres in capybaras were significantly higher for R. bellii, according to Luz et al. (2019). My interest would be in the response regarding the behaviour of both the host reservoir and the Amblyomma sculptum tick population in this group “R. bellii-infected guinea pig via intraperitoneal route”, which was not carried out in this experiment. Is there any reason not to include this group in the experiment, even if it does not mimic what happens in the natural environment, besides the clarification related to the work of Pacheco et al. (2007)? As also discussed in the manuscript, capybaras could ingest R. bellii-infected ticks while feeding and for this reason would have anti-R. bellii antibodies. I apologise if I was not clear in these questions: I just wanted to understand the absence of this group in the experiment.

Another issue that I consider significant is that the authors reported that the guinea pigs were exposed to R. bellii because found this rickettsia in the salivary gland A. dubitatum tick, ok? The question is: were the animals really exposed to R. bellii, since they did not develop anti-R. bellii antibodies using the IFA technique? And if there is any “unknown factors inherent to A. dubitatum tick” that prevents the entry of this bacterium into guinea pigs during the tick’s blood meal? In this case, it is likely that the guinea pigs were not exposed to R. bellii, right?

Materials and Methods

Pag. 5; line 100: If possible, change “Ticks” to “Animals and Ticks”

Results

Pag. 11; lines 257-258: “These coupled results suggest that animals from the GB group were in fact exposed to R. bellii in spite of the absence of a humoral host response”. Please, review the work of Sakay et al. (2014) which states that “These results suggest that R. bellii was not tick-transmitted to CG guinea pigs. Guinea pigs exposed to ticks infected by both R. rickettsii and R. bellii generally elicited endpoint titers much higher to R. rickettsii than to R. bellii, indicating that reactions to the later agent were a result of cross-reactions with anti-R. rickettsii antibodies”. The question is: Does R. bellii-infected A. dubitatum tick transmit the bacterium to guinea pigs or not?

Pag. 11; lines 273-275: In this way, we can consider that all animals in the GB group were not exposed to rickettsia, correct?

Discussion

Pag. 11; lines 331-332: Or, for some unknown reason, the tick failed to transmit the bacteria during the blood meal, even though it was detected by direct methods in the salivary glands.

REVIEWER #2

The article by Binder and colleagues describes a study aimed at exploring an interesting phenomenon regarding the ecology of Rickettsia rickettsii in Brazil, where Rocky Mountain spotted fever (also known as Brazilian spotted fever) is endemic. In general, the manuscript is well written.

The overall experimental design is good, but this design is greatly limited by the very few animals. Although the total number of ticks is great, the number of animals for which the experiment was conducted was small (10 guinea pigs separated into 3 experimental groups) with lack of justification for these small numbers via power analysis. As an outbred animal model, one can expect more variation limiting validity of results with the few animals used. In addition, one animal (a control) was not analyzed due to the assumption that it was not infected (it should be noted that even ~ 12-15% of Hartley strain guinea pigs are completely resistant to infection with Rickettsia rickettsii). Although there is a statistical difference between the infection rate of nymphs, these nymphs were collected from very few animals, which questions if the comparison of these results to the situation in nature is valid.

The length of the manuscript is not commensurate for the amount and type of data presented. Both the introduction, discussion, and reference lists should be consolidated.

Lines 175 – 176: Please provide the dilution of the secondary antibody.

Line 180: Comment. A dilution of 1:64 is quite low for a positive control. Would usually expect a known positive to require a higher dilution for optimized viewing. If indeed this low dilution was required, it brings to question the quality of antigen used for the IFA analysis.

Llines 338 – 339: Seropositivity through ingestion of Rickettsia-infected ticks rather than tick bites? Seems unlikely.

## AUTHORS’ RESPONSE TO THE REVIEWERS

We thank the reviewers for their valuable suggestions and important criticisms. Below are the responses to each of the comments.

We would also like reviewers to be aware that we conducted an additional experiment to address some of the questions raised, which is described in the new version of the manuscript (L. 237 – 262; L. 285 - 289).

Reviewer:1

The manuscript titled “Effects of exposure to Rickettsia bellii on reservoir competence of guinea pigs for Rickettsia rickettsii” contains relevant information, and I recommend it for publication. However, I suggest that the authors make some minor changes, if they deem it appropriate.

After reading about the experimental design, I felt that an experimental group exposed to R. bellii through intraperitoneal inoculation was missing, so that we would have a group with anti-R. bellii antibodies, since in non-endemic areas of BSF, the final titres in capybaras were significantly higher for R. bellii, according to Luz et al. (2019). My interest would be in the response regarding the behaviour of both the host reservoir and the Amblyomma sculptum tick population in this group “R. bellii-infected guinea pig via intraperitoneal route”, which was not carried out in this experiment. Is there any reason not to include this group in the experiment, even if it does not mimic what happens in the natural environment, besides the clarification related to the work of Pacheco et al. (2007)? As also discussed in the manuscript, capybaras could ingest R. bellii-infected ticks while feeding and for this reason would have anti-R. bellii antibodies. I apologise if I was not clear in these questions: I just wanted to understand the absence of this group in the experiment.

RESPONSE: Thanks for this interesting and relevant comment. Indeed, if we were to repeat this experiment today, we would definitely include a group with intraperitoneal inoculation of R. bellii. However, at the time when we planned this study, we were interested in mimicking a natural exposure to R. bellii via tick bite, because we thought that this would be the most common natural infection route. At that time, we did not imagine that guinea pigs would not develop a humoral response against R. bellii. Although it was published earlier, the other study in which our team observed this behavior (Binder et al., 2025) was carried out several months after this one. That is why we did not include other experimental groups in the study (e.g. a group exposed to R. bellii via tick ingestion).

Another issue that I consider significant is that the authors reported that the guinea pigs were exposed to R. bellii because found this rickettsia in the salivary gland A. dubitatum tick, ok? The question is: were the animals really exposed to R. bellii, since they did not develop anti-R. bellii antibodies using the IFA technique? And if there is any “unknown factors inherent to A. dubitatum tick” that prevents the entry of this bacterium into guinea pigs during the tick’s blood meal? In this case, it is likely that the guinea pigs were not exposed to R. bellii, right?

RESPONSE: We thought that detecting R. bellii in salivary glands was enough to state that this bacterium was inoculated in guinea pigs, but after this valuable questioning, we decided to verify whether R. bellii is in fact secreted in the saliva. Thus, we conducted an additional experiment, which is described in the new version of this manuscript, in which we were able to detect R. bellii in the saliva of infected A. dubitatum, corroborating the hypothesis that guinea pigs were indeed exposed to R. bellii, despite of the absence of detectable humoral response (L. 237 – 262; L. 285 - 289).

Materials and Methods

Pag. 5; line 100: If possible, change “Ticks” to “Animals and Ticks”

RESPONSE: The word “animals” was added.

Results

Pag. 11; lines 257-258: “These coupled results suggest that animals from the GB group were in fact exposed to R. bellii in spite of the absence of a humoral host response”. Please, review the work of Sakay et al. (2014) which states that “These results suggest that R. bellii was not tick-transmitted to CG guinea pigs. Guinea pigs exposed to ticks infected by both R. rickettsii and R. bellii generally elicited endpoint titers much higher to R. rickettsii than to R. bellii, indicating that reactions to the later agent were a result of cross-reactions with anti-R. rickettsii antibodies”. The question is: Does R. bellii-infected A. dubitatum tick transmit the bacterium to guinea pigs or not?

RESPONSE: Sakai et al (2014) did not attempt to detect R. bellli neither in the salivary glands or in the saliva of A. dubitatum, so, as there was no detectable humoral response, they assumed that R. bellii was not transmitted to guinea pigs. In this study we were able to better investigate the behavior of R. bellii in A. dubitatum ticks and it seems very unlikely that R. bellii is not transmitted to guinea pigs, since the bacterium was detected in salivary glands and saliva in 100% of the ticks that were tested.

Pag. 11; lines 273-275: In this way, we can consider that all animals in the GB group were not exposed to rickettsia, correct?

RESPONSE: In this paragraph we are referring to the absence of humoral response and clinical manifestations in animal C31 after exposure to R. rickettsii. All animals of the other groups had clinical signs and seroconverted after exposure to R. rickettsii. The absence of humoral response after exposure to R. bellii is discussed along the manuscript.

Discussion

Pag. 11; lines 331-332: Or, for some unknown reason, the tick failed to transmit the bacteria during the blood meal, even though it was detected by direct methods in the salivary glands.

RESPONSE: Since we were able to show that A. dubitatum secretes R. bellii in the saliva, we do not see a plausible reason for this bacterium not being transmitted to guinea pigs. Additionally, our findings suggest a different immunological behavior among guinea pigs previously exposed to R. bellii, which reinforces that these animals were indeed infected by this bacterium.

Reviewer 2:

The article by Binder and colleagues describes a study aimed at exploring an interesting phenomenon regarding the ecology of Rickettsia rickettsii in Brazil, where Rocky Mountain spotted fever (also known as Brazilian spotted fever) is endemic. In general, the manuscript is well written.

The overall experimental design is good, but this design is greatly limited by the very few animals. Although the total number of ticks is great, the number of animals for which the experiment was conducted was small (10 guinea pigs separated into 3 experimental groups) with lack of justification for these small numbers via power analysis. As an outbred animal model, one can expect more variation limiting validity of results with the few animals used. In addition, one animal (a control) was not analyzed due to the assumption that it was not infected (it should be noted that even ~ 12-15% of Hartley strain guinea pigs are completely resistant to infection with Rickettsia rickettsii). Although there is a statistical difference between the infection rate of nymphs, these nymphs were collected from very few animals, which questions if the comparison of these results to the situation in nature is valid.

RESPONSE: The small number of animals is unfortunately a limitation of the study. It is difficult nowadays to justify using a larger number of animals for this type of experiment. Rivas et al. (2015), for example, investigated protective immunity against R. rickettsii elicited by R. amblyommatis using only eight guinea pigs, while Levin et al. (2014) assessed effects of heterologous immunization in reservoir competence of dogs using only six animals. Furthermore, infection by R. rickettsii causes substantial distress in guinea pigs and this was a fairly long experiment, with animals being kept in individual cages without social interaction for several months, what makes even more difficult to advocate for the use of larger numbers of animals.

The length of the manuscript is not commensurate for the amount and type of data presented. Both the introduction, discussion, and reference lists should be consolidated.

RESPONSE: Reference list was consolidated. On the other hand, the Introduction was formulated to contextualize the application of the experiments used in this study, while the Discussion section discusses precisely the application of these results in the ecology of Brazilian spotted fever. Unfortunately, we were unable to determine what could be omitted as being of lesser importance. If this reviewer can suggest this in a more objective manner, we would be grateful.

Lines 175 – 176: Please provide the dilution of the secondary antibody.

RESPONSE: Dilution of the secondary antibody is now described.

Line 180: Comment. A dilution of 1:64 is quite low for a positive control. Would usually expect a known positive to require a higher dilution for optimized viewing. If indeed this low dilution was required, it brings to question the quality of antigen used for the IFA analysis.

RESPONSE: Although we used a dilution of 1:64 for the positive control serum, this serum had an endpoint titer higher than 1024, as obtained from guinea pigs previously inoculated with R. rickettsii in our laboratory. We have added the reference of Gerardi et al. (2019) for this positive control serum in the text. We used a dilution of 1:64 just because it was the cut-off dilution for the test serum samples. But indeed, this same positive control serum could be used at higher dilutions (e.g., 1:512 or 1:1024) without any interference in our serological analyses.

Lines 338 – 339: Seropositivity through ingestion of Rickettsia-infected ticks rather than tick bites? Seems unlikely

RESPONSE: It is just a hypothesis, not a conclusion or a statement.

---

## [Reviewer Report · REVIEWERS COMMENTS]

## REVIEWER #1

The authors included a microscopy image of tick saliva to demonstrate the presence of bacteria (R. belli) and, consequently, transmission, but they do not mention the bacterial load and viability in the saliva (I found it quite low—only two bacteria in the image). Were the bacteria viable and actually capable of establishing an infection and, “somehow,” preventing infection by R. rickettsii? Or is referee 2 correct in mentioning that resistance to infection can occur in these animals?

REVIEWER #2

The manuscript addresses this reviewer’s minor comments, but it is unresponsive to comments regarding animal numbers. Their justification based on other small studies with different experimental designs is not an appropriate approach to study design. Distress is here used as a convenient rationale, rather than a scientifically based one. No additional discussion about these weaknesses of low animal numbers is provided in the discussion. At the very least, the latter is required.

## AUTHORS’ RESPONSE TO THE REVIEWERS

We thank the reviewers for their valuable suggestions and important criticisms. Below are the responses to each of the comments.

Reviewer 1:

The authors included a microscopy image of tick saliva to demonstrate the presence of bacteria (R. belli) and, consequently, transmission, but they do not mention the bacterial load and viability in the saliva (I found it quite low—only two bacteria in the image). Were the bacteria viable and actually capable of establishing an infection and, “somehow,” preventing infection by R. rickettsii? Or is referee 2 correct in mentioning that resistance to infection can occur in these animals?

Response:

In the previous version of the manuscript, this reviewer had made the following question: “In this case, it is likely that the guinea pigs were not exposed to R. bellii, right?” With our new procedure to demonstrate R. bellii organisms in the saliva of A. dubitatum, we can confirm that guinea pigs were exposed to R. bellii after been infested with ticks naturally infected with this agent. Since these A. dubitatum ticks were naturally infected with R. bellii, and were shown to maintain this infection through successive life stages and generations in our laboratory, we see no reasonable evidence that the bacteria were not viable.

Regarding our new figure of the manuscript, we see two bacteria at a magnification of 1000x in a single field. In 1µl of saliva (volume of saliva observed under the microscope), we observed more than 20 bacteria in all samples. Considering that ticks salivate for days on the host, we do not believe this to be a negligible amount of bacteria. We are unaware of any studies that have quantified Rickettsia in tick saliva, so any results we obtained would not be comparable to other studies or other Rickettsia species. Regardless, we recall that the procedure for demonstrating the presence of rickettsia in saliva was to answer this reviewer’s final question: “In this case, it is likely that the guinea pigs were not exposed to R. bellii, right?”. Obviously, quantification of rickettsia (bacteria load) in tick saliva is far beyond of the scope of this manuscript.

Please note that in our Discussion, we do not argue that R. bellii was able to establish an infection in the host, but only that it could have induced a cellular immune response, based on the general lower mortality of guinea pigs after being exposed to R. bellii-infected ticks. We would like to reinforce that guinea pigs are the animal model of choice for R. rickettsii infection, and we are unaware of any studies showing that any of these animals are resistant to highly virulent strains of R. rickettsii. Just to point out, we used a strain of R. rickettsii that is very virulent for R. rickettsii, as demonstrated in previous studies of experimental infection with this strain in guinea pigs (Soares et al. 2012, Gerardi et al. 2019).

Reviewer 2:

The manuscript addresses this reviewer’s minor comments, but it is unresponsive to comments regarding animal numbers. Their justification based on other small studies with different experimental designs is not an appropriate approach to study design. Distress is here used as a convenient rationale, rather than a scientifically based one. No additional discussion about these weaknesses of low animal numbers is provided in the discussion. At the very least, the latter is required.

Response: A discussion about limitations of using a small sample of animals was included in the manuscript (L 411 - 414).